# Grain Size and Phase Transformation Behavior of TiNi Shape-Memory-Alloy Thin Film under Different Deposition Conditions

**DOI:** 10.3390/ma13143229

**Published:** 2020-07-20

**Authors:** Joohyeon Bae, Hyunsuk Lee, Duckhyeon Seo, Sangdu Yun, Jeonghyeon Yang, Sunchul Huh, Hyomin Jeong, Jungpil Noh

**Affiliations:** 1Department of Energy and Mechanical Engineering and Institute of Marine Industry, Gyeongsang National University, 2 Tongyeonghaean-ro, Tongyeong 53064, Korea; jhyeonb@gnu.ac.kr (J.B.); no9192@gnu.ac.kr (H.L.); dudug17@gnu.ac.kr (D.S.); schuh@gnu.ac.kr (S.H.); hmjeong@gnu.ac.kr (H.J.); 2Department of Mechanical System Engineering, Gyeongsang National University, 2 Tongyeonghaean-ro, Tongyeong 53064, Korea; sdyun@gnu.ac.kr (S.Y.); jh.yang@gnu.ac.kr (J.Y.)

**Keywords:** TiNi thin film, shape memory alloy, grain size, phase transformation behavior

## Abstract

TiNi shape-memory-alloy thin films can be used as small high-speed actuators or sensors because they exhibit a rapid response rate. In recent years, the transformation temperature of these films, manufactured via a magnetron sputtering method, was found to be lower than that of the bulk alloys owing to the small size of the grain. In this study, deposition conditions (growth rate, film thickness, and substrate temperature) affecting the grain size of thin films were investigated. The grain size of the thin film alloys was found to be most responsive to the substrate temperature.

## 1. Introduction

Shape memory alloys (SMAs) are commonly considered to be smart materials because they exhibit shape memory effects and hyper-elastic properties that are not observed in typical metals [1,2,3,4,5,6]. These extraordinary properties are due to a temperature-dependent martensitic phase transformation from a low (martensite) to high (austenite) symmetric crystallographic structure upon cooling, and a reverse martensitic transformation in the opposite direction upon heating.

Based on the constituent elements, these materials are classified into TiNi-, Cu-, and Fe-based alloys, among which TiNi-based shape memory alloys are the most widely used owing to their excellent stability and corrosion resistance [7,8,9,10,11]. TiNi-based shape memory alloys are either manufactured in bulk or in a thin film form. The SMA thin films have many desirable properties, including high power density (up to 10 J/cm^3^), a fast response time, the ability to recover large transformation stress and strain upon heating, shape memory effect, peudoelasticity, and biocompatibility [12,13,14,15]. In particular, a fast response time is essential for the development of high-speed actuators or sensors.

Methods such as pulsed laser deposition (PLD), cathodic arc ion plating (CAIP), and sputtering are used to fabricate the SMA thin films [16,17]. Sputtering deposition, which is easily applicable to industries, is the most studied among these techniques [18,19,20]. Recently, a TiNi shape memory alloy thin film was fabricated via a direct current (DC) magnetron sputtering method, and the crystallization and phase transition behaviors of the thin film were investigated. The thin film grown at room temperature was amorphous and was capable of crystallizing through annealing at 873 K. However, its phase transformation temperature was much lower (below 123 K) than that of the bulk alloy with a similar composition, and this result was found to be attributable to its small grain size [21].

This study investigated the variations in the grain size and phase transformation characteristics of TiNi shape memory alloy thin films prepared using DC magnetron sputtering, under different deposition conditions.

## 2. Materials and Methods

### 2.1. Growth of the Thin Film

A DC magnetron sputtering method was used to create TiNi shape memory alloy thin films. Glass with a thickness of 1 mm, width of 26 mm, and length of 76 mm was used as the substrate, and ultrasonic cleaning was performed using acetone and alcohol for 10 min each. The diameter of the Ti and Ni targets used for deposition was 2 inches, and the purity levels were 99.995% and 99.99%, respectively. An initial vacuum of 6.67 × 10^−4^ Pa was maintained using a rotary pump (Gyeongsan, Gyeongbuk, Korea) and turbo-molecular pump (OSAKA VACUUM, Osaka, Japan), and deposition was performed in an argon atmosphere at 0.199 Pa. The distance between the target and the substrate during deposition was 80 mm, and the substrate was rotated at a speed of 30 rpm to obtain a uniform thin film composition over the entire substrate. Pre-sputtering was conducted for 10 min to remove impurities on the target surface, and the power setting for actual thin film growth was adjusted within the range of 45–540 W for Ti and 15–186 W for Ni. The temperature level of the substrate was changed from 293 K to 773 K, and the deposition time was adjusted from 30 min to 480 min. Analyses on the cross-section and composition of the deposited thin film revealed that the thickness of the thin film was in the range 1.6–9.3 μm and the composition was Ti-50 ± 0.3 Ni (at %). Upon separation from the substrate, the thin film was vacuum sealed in a quartz tube and annealed for crystallization. During this process, the vacuum was set at 2.7 × 10^−3^ Pa, and annealing was conducted at 873 K for 1 h.

### 2.2. Characterization of Thin Films

Compositional analysis, cross-sectional observation, and surface morphology characterization of the prepared thin film was performed using field-emission scanning electron microscopy (FE-SEM, and MIRA3 LM, TESCAN, Brno-Kohoutocice, Czech Republic). The crystal structure of the thin film was analyzed using an X-ray diffractometer (XRD, Bruker AXS, D2 PHASER, BRUKER, Billerica, MA, USA). XRD analysis was performed at a scan speed of 0.016/s over a 2θ range of 30–80°. To characterize the phase transformation behavior, differential scanning calorimetry (TA Instrument, DSC Q20, New Castle, DE, USA) and low temperature XRD (Bruker AXS, D8 Advancer) experiments were conducted. DSC measurements involved cooling and heating within the range 423–123 K, and the cooling/heating rate was set at 0.166 K/s. DSC specimens with masses between 10 and 15 mg were heated from room temperature up to 423 K where they were held for five minutes to establish thermal equilibrium. Then, the DSC measurements were recorded upon cooling to 123 K at 0.166 K/s. At 123 K the specimen was held for five minutes and then heated to 423 K with a heating range of 0.166 K/s. The average grain size was measured using the linear intercept method. As per the recommendation of ASTM E1382, four directions (0°, 45°, 90°, 135°) were used [21]. Ten lines were drawn in each direction to measure, and the average value was calculated and used.

## 3. Results

Figure 1 presents the surface morphology, XRD, and DSC results of Ti-49.8Ni (at %) thin film annealed at 873 K for 1 h. The grain size of the annealed thin film was approximately 29 nm as shown in Figure 1a. XRD analysis (Figure 1b) indicated that the annealed thin film has a B2 (cubic) structure, and DSC analysis (Figure 1c) showed no clear signs of phase transformation within the 123–423 K temperature range [22]. These results are in good agreement with those from previous studies, which determined that the martensite phase transformation temperature for grain sizes below 50 nm dropped to 100 K [23].

Figure 2 displays the results obtained from an investigation on the change in grain size based on the controlled growth rate of the thin film. The growth rate was determined by controlling the power setting of the Ti and Ni targets, resulting in thin film growth rate changes from 4.64 to 56.67 nm/min. The power of the Ti and Ni targets during deposition was adjusted from 45 to 540 W and 15 to 186 W, respectively. The final thickness of the thin film grown under each power setting was 1.69 ± 0.2 μm, and the composition obtained from energy dispersive X-ray spectroscopy (EDS) analysis was Ti-50.0 ± 0.3 Ni (at %). As shown in Figure 2a–e, the grain size changed to 16.6, 19.0, 20.1, 20.8, and 20.2 nm; as the growth rate increased. The grain size increased until the growth rate reached 38.0 nm/min, beyond which no increase in grain size was observed (Figure 2f). As the power of the target increased, the incident ionic energy also increased. Hence, it is speculated that the increase in grain size is a result of the enhanced mobility of the atoms adsorbed to the substrate when particles with high energy collide with the surface [24].

Figure 3 shows the SEM images of TiNi thin films with different film thickness values. The deposition times in Figure 3a–d are 90, 270, 360, and 480 min, respectively. The deposition power was fixed at 180 W for Ti and 62 W for Ni. As the deposition time increased, the grain size increased in the order of 20.1, 25.7, 28.4, and 30.5 nm. This increase in grain size with increasing thickness of the thin film is attributed to the growth of grains with orientations that minimize the surface energy (to minimize the energy of the entire system) at the expense of grains with different orientation [25,26]. However, a thin film is by definition a layer of material with a thickness that is below several microns, and therefore, controlling the film thickness to increase the grain size is considered to be disadvantageous.

Figure 4 displays images obtained from the observation of TiNi thin film surfaces at different substrate temperatures. The substrate temperatures shown in Figure 4a–d are 293, 623, 723, and 773 K, respectively. The power during deposition was set to 180 W for Ti and 62 W for Ni, and the deposition time was 270 min. As the substrate temperature increased from 293 to 773 K, grain sizes of 25.7, 28.3, 31.9, and 214 nm were observed. As demonstrated in Figure 4e, the grain size gradually increased as the substrate temperature increased. However, the grain size of the thin film grown at 773 K increased drastically when compared with those observed at lower substrate temperatures. This drastic increase in the grain size at a specific temperature occurred because the energy of the sputtered atoms exceeded the activation energy of inter-diffusion, causing active inter-diffusion [27]. Therefore, these experiments demonstrate that the substrate temperature is the most important factor dictating the grain size.

## 4. Discussion

Figure 5 shows the DSC results of TiNi alloy thin films grown at 723 K and 773 K. No distinct peaks were observed during the cooling and heating processes of the thin film grown at 723 K (Figure 5a). That is, phase transformation did not lead to exothermic and endothermic reactions. However, the thin film grown at 773 K showed two events that corresponded to an exothermic reaction during cooling and one endothermic reaction during heating (Figure 5b). Transformation temperature of R_s_ (the R phase transformation start temperature), M_s_ (the B19′ phase transformation start temperature), M_f_ (the B19′ phase transformation finish temperature), A_s_ (the reverse transformation start temperature), and A_f_ (the reverse transformation finish temperature) were 294, 238, 202, 268, and 295 K, respectively. It is thought that the reason only one peak is observed in the heating process is because the A_s_ temperature is lower and overlaps with the reverse R phase transformation (B19′→R). It has been reported that in the heat-treated TiNi alloy, A_s_ temperature is greatly decreased by the annealing condition, and the two transformation temperatures (B19′→R and R→B2) appear to overlap [28,29,30].

The crystalline structure was analyzed by conducting low temperature XRD measurements aimed at accurately determining the phase transformation behavior with respect to temperature change. Figure 6 shows the XRD results from different temperature conditions.

XRD measurements were conducted by cooling and heating the temperature from 363 to 183 K, at a scan speed of 0.016°/s over a 2θ range of 35–50°. The temperature deviation of each set temperature was ±0.5 K. As illustrated in Figure 6a, the thin film grown at 723 K, exhibited B2 structure at all measured temperatures. This result indicated that a martensite phase transformation did not occur when the crystallized thin film was cooled to 183 K, which is consistent with the DSC results in Figure 5a. It is also consistent with a previous finding that demonstrates the transformation temperature falls below 123 K due to the small grain size [23]. In contrast, the thin film grown at 773 K showed signs of phase transformation as the temperature changed (Figure 6b,c). Figure 6b shows the XRD results measured during the cooling process. The B2 structure was observed at 323 K, and the B2 phase transformed to the rhombohedral (R) phase when the temperature was decreased to 275 K. Decreasing the temperature further led to the transition of the R phase to B19′ (monoclinic) phase. As such, the first peak observed during cooling in the DSC experiment shown in Figure 5b corresponds to the transformation of B2 to R, and the second peak corresponds to the transformation of the R to the B19′ phase. Figure 6c shows the XRD results during the heating process. The B19′ phase was observed at 230 K, and the R phase and B19′ phase were observed simultaneously at 279 and 290 K. When the temperature was heated to 295 K, the R phase and B19′ phase transformed to the B2 phase. Although only one peak was observed in the DSC heating curve in Figure 5b, it was confirmed that a two-step transformation behavior (B19′→R→B2) occurred during the heating process. Thus far, Ti-50Ni (at %) bulk alloys have been known to exhibit a single-step B2 to B19′ phase transformation behavior [31]. However, when prepared as thin films, these materials were shown to undergo a two-step process involving the phase transformation of B2 to R and subsequently to B19′. It is postulated that exploiting the small transformation history of the R-phase transition region would produce actuators with faster response times compared to those made from bulk alloys. In addition, the DSC (Figure 5) and low temperature XRD (Figure 6) results confirmed that the change in phase transformation temperature is sensitive to the grain size.

## 5. Conclusions

DC magnetron sputtering was used to investigate the grain size and phase transformation behavior of TiNi shape memory alloy thin films under different deposition conditions. Alloy thin films grown at low substrate temperatures were amorphous and capable of crystallizing through annealing at 873 K. However, the phase transition temperature dropped below 123 K due to the small grain sizes. Increasing the growth rate led to an increase in the grain size of the thin film. However, the increase in the grain size seized when the growth rate exceeded 38.0 nm/min. An increase in the thickness of the thin film from 1.69 to 9.35 nm did not result in a significant increase in grain size (20.1→30.5 nm). A drastic increase in the grain size from 31.0 to 214 nm occurred when the substrate temperature was increased from 723 to 773 K. This observation revealed that the substrate temperature had the most significant effect on the grain size. The transformation temperatures of TiNi shape memory alloy thin films with identical elemental composition deposited at 773 K were R_s_ = 298 K, M_s_ = 238 K and A_s_ = 268 K. In addition, the two-step phase transformation behavior exhibited in B2→R→B19′ was established from low temperature XRD measurements. Furthermore, the change in transformation temperature was found to be very responsive to the grain size.

## Figures and Tables

**Figure 1 materials-13-03229-f001:**
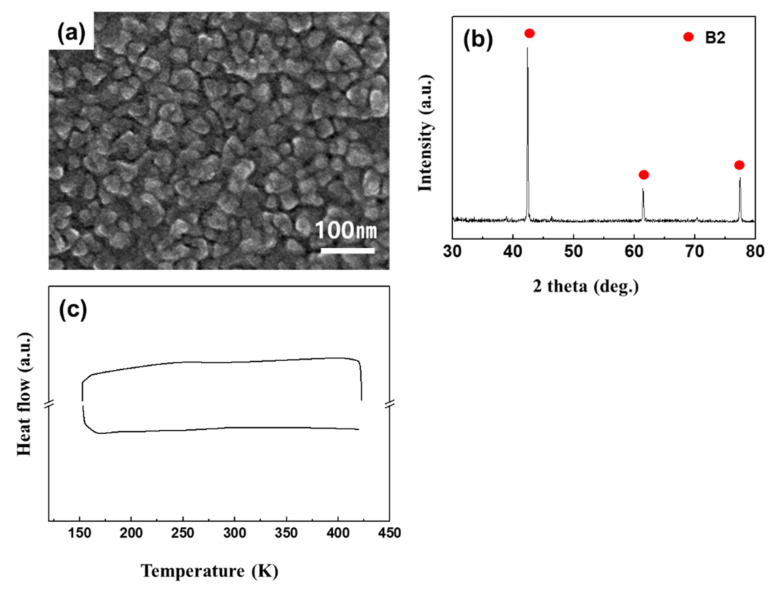
(**a**) Surface image, (**b**) XRD and (**c**) DSC results of Ti-49.8Ni (at %) thin film annealed at 873 K for 1 h. (Deposition condition: growth rate (6.29 nm/min), thickness (1.7 μm), substrate temperature (293 K)).

**Figure 2 materials-13-03229-f002:**
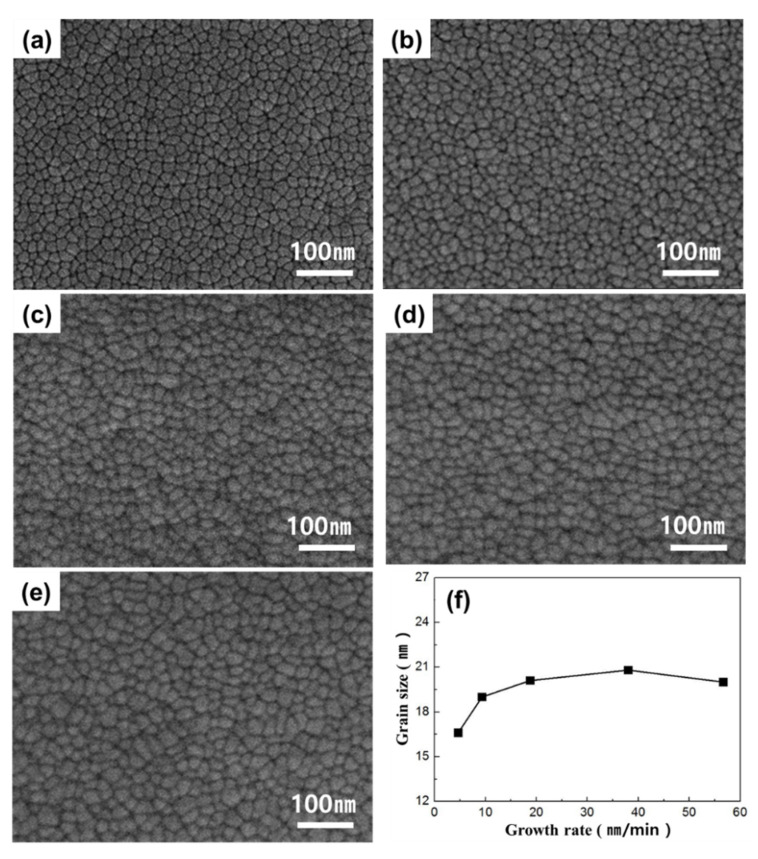
FE-SEM images of Ti-50.0 ± 0.3Ni (at %) thin film with growth rates of (**a**) 4.64 nm/min, (**b**) 9.33 nm/min, (**c**) 18.78 nm/min, (**d**) 38.00 nm/min, and (**e**) 56.67 nm/min; and (**f**) relationship between grain size and growth rate. (Deposition condition: thickness (1.69 ± 0.2 μm), substrate temperature (293 K)).

**Figure 3 materials-13-03229-f003:**
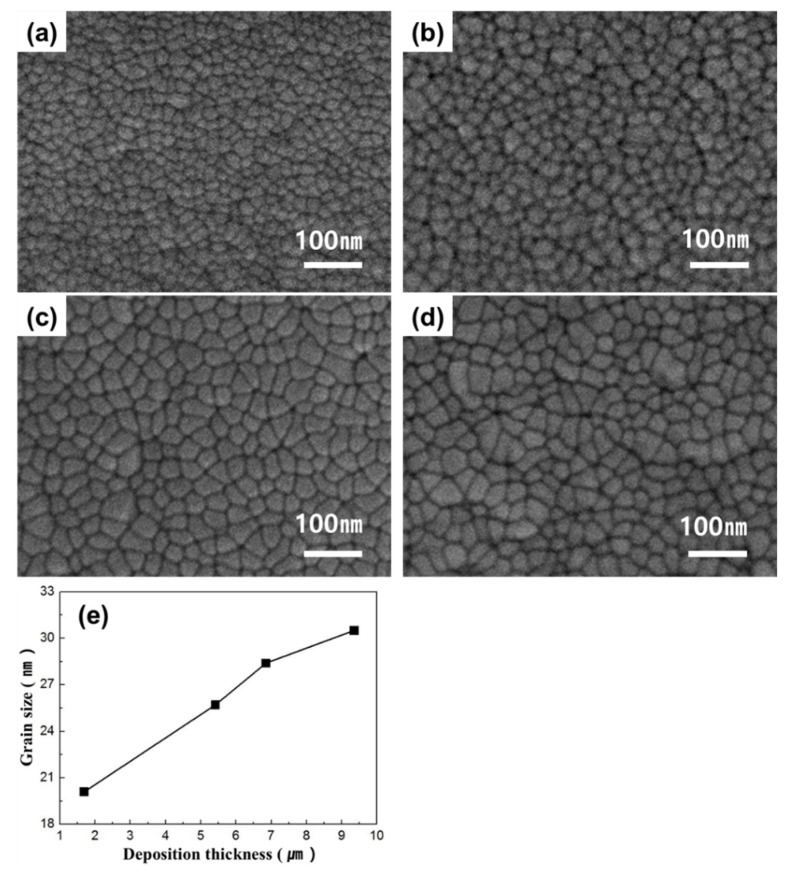
FE-SEM images of Ti-50.0 ± 0.3Ni (at %) thin film with film thickness values of (**a**) 1.69 μm, (**b**) 5.41 μm, (**c**) 6.85 μm, and (**d**) 9.35 μm; and (**e**) relationship between grain size and film thickness. (Deposition condition: growth rate (6.29 nm/min), substrate temperature (293 K)).

**Figure 4 materials-13-03229-f004:**
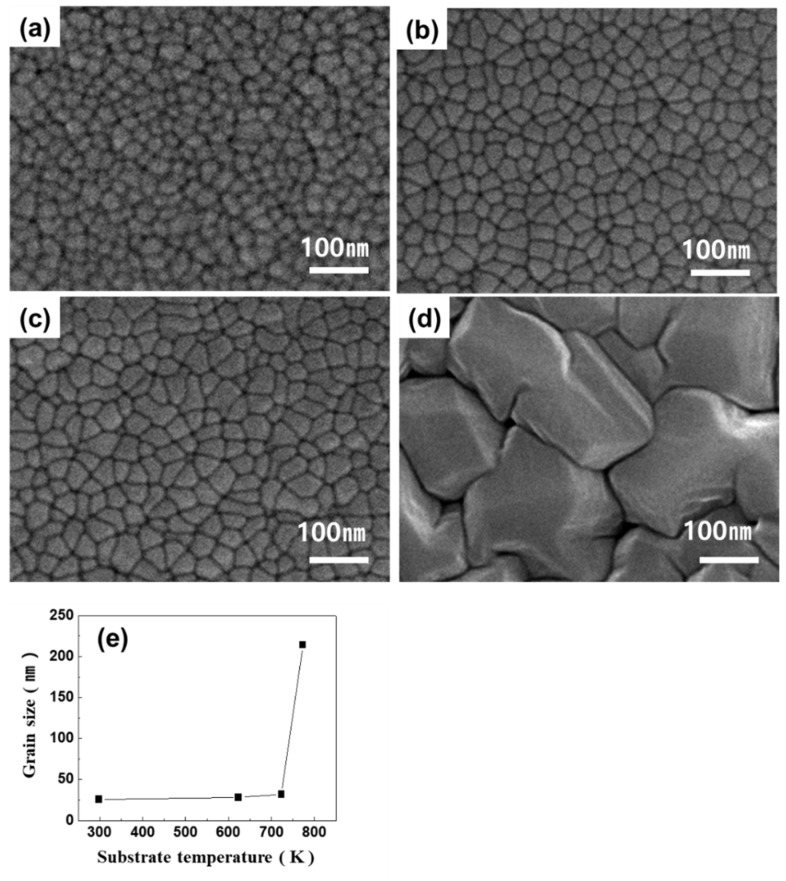
FE-SEM images of Ti-50.0 ± 0.3Ni (at %) thin film with substrate temperatures of (**a**) 293 K, (**b**) 623 K, (**c**) 723 K, and (**d**) 773 K; and (**e**) relationship between grain size and substrate temperatures. (Deposition condition: growth rate (6.29 nm/min), thickness (1.7 μm)).

**Figure 5 materials-13-03229-f005:**
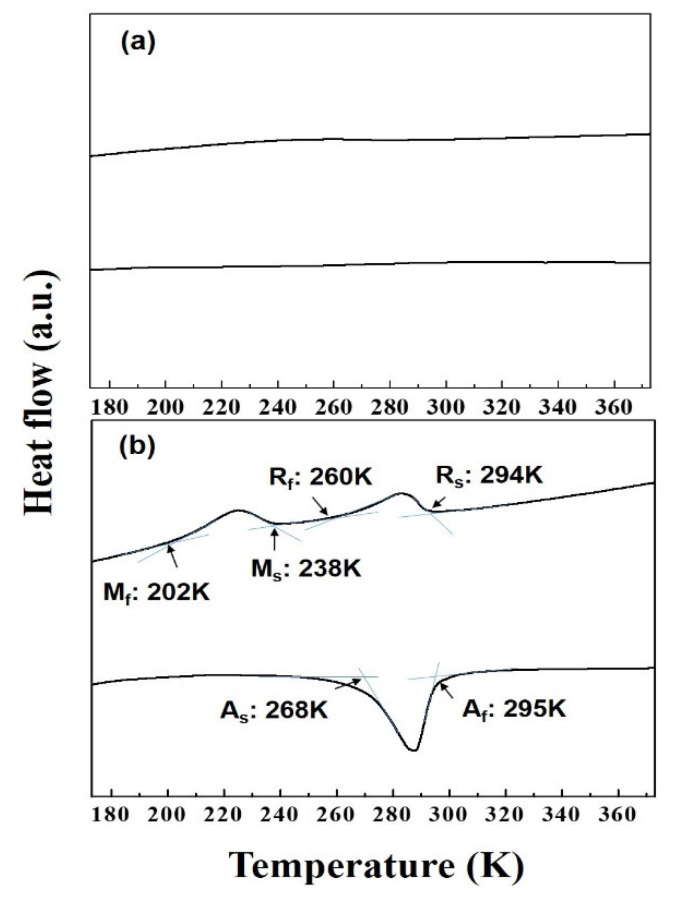
DSC results of Ti-49.9Ni (at %) thin film with a substrate temperature of (**a**) 723 K and (**b**) 773 K. (Deposition condition: growth rate (6.29 nm/min), thickness (1.7 μm)).

**Figure 6 materials-13-03229-f006:**
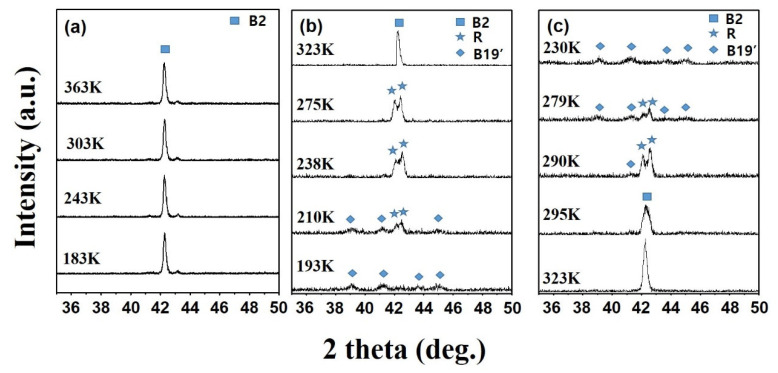
Low temperature XRD results of Ti-49.9Ni (at %) thin film with a substrate temperature of (**a**) 723 K, (**b**) 773 K (cooling,) and (**c**) 773K (heating). (Deposition condition: growth rate (6.29 nm/min), thickness (1.7 μm)).

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
