# Peer review of "Grain Size and Phase Transformation Behavior of TiNi Shape-Memory-Alloy Thin Film under Different Deposition Conditions"

_materials, 2020, doi:10.3390/ma13143229_

Round 1

Reviewer 1 Report

Materials-856490

Grain size and phase transformation behaviour of TiNi shape-memory-alloy thin film under different deposition conditions

by J. Bae et al.

The authors of the above-mentioned manuscript investigate the grain size and the phase formation of TiNi shape-memory-ally thin films which is definitely an interesting topic for the readership of “Materials”. The influence of film thickness, deposition rate, as well as the substrate temperature is investigated and described. The results are really interesting nevertheless they are partly not comprehensible and not properly presented. After a major revision of the manuscript and the presented data the manuscript can definitely be considered for publication.

Comments:

  1. The manuscript should be thoroughly revised regarding the language and typing errors there are several mistakes distributed in the whole manuscript.

  1. The authors mention the term “deposition thickness” several times. I think that is not a proper term more suitable would be “film thickness” or “coating thickness”.

  1. Figure 1 shows results of an annealed sample which was deposited at room temperature. This figure should be presented after presenting the as deposited samples (which would be Figure 2. Furthermore, the authors state that the coating investigated exhibits a growth rate of 6.29 nm/min in Figure 2 (same substrate temperature and different growth rates) no sample with a growth rate of 6.29 nm/min is shown, why? The authors should comment on the that, respectively show the sample as for all the investigations the deposition rate of 6.29 nm/min is mentioned.
    1. The authors mention a grain size of 29 nm for the sample annealed at 873 K. What is the initial grain size? As the grain size for the coating deposited at 773 K is 214 nm (shown in Fig 4d) I would expect significant grain growth upon annealing the sample deposited at 273 K to 873 K, can the authors comment on that?

  1. According to the experimental part it is not possible to follow the DSC measurements. I assume the starting condition of the DSC is room temperature!? What is the heating/cooling sequence? As the measurements (Fig 1 and Fig 5) do not start at room temperature it would be interesting if the sequence is coolingà starting measurement àheatingàcooling or heatingàstarting measurementàcoolingàheating?
    1. In Fig 5 the authors label the y-Axis with “Heat flow (W/g)” in Fig 1 with “Heat flow (a.u.)” this should be made consistently throughout the manuscript.
    2. The same applies for the XRDs in Fig 1 the y-axis is labelled Intensity (a.u.) and the x-axis 2 theta (deg.) in Fig. 6 there is no labelling of the axes.
    3. Is the intensity of the XRDs on a linear scale or is it logarithmic as it is also often seen?

  1. The authors state that the thin films deposited at room temperature are amorphous and crystallise upon annealing up to 873 K. An XRD measurement of the as deposited sample should be shown to prove that the thin film is amorphous. What is the actual crystallisation temperature of the coatings? Coatings deposited at 723 K show B2 structure are the coatings deposited with 623 K crystalline?

  1. How is the grain size determined? I am not sure if a difference of 0.7 nm can be determined to really be a difference. The authors should mention how they determine the grain size and may also give a standard deviation.

 In this regard I do not think it is appropriate to give two decimal digits for the growth rate (in nm/min) which would be the order of 10 pm if the film thickness shows a deviation of 200 nm (e.g. mentioned in line 82: “1.69±2 µm”)

  1. Figure 5 shows the DSC signals for heating and cooling of the samples deposited at 723 K and 773 K respectively. The authors mention that the phase transformation upon cooling occurs via a two-step mechanism B2àRàB19’. Why does the phase transformation upon heating occur via one step? The authors do not explain this behaviour.

  1. In Fig 6 the authors present the XRD measurements carried out at different temperatures. Why do the authors choose different temperatures for the different samples?

 For Fig. 6a the authors mention a single phase B2 structure. What is about the distinct peak slightly above 43°? It seems that the coating does not exhibit a single-phase structure. Can the authors comment on that?

 In Fig 6b the mentioned phase transformations can be observed. What is the initial structure after deposition? What is the cooling/heating sequence for the XRD measurements? I am missing some details to these experiments.

Basically, the results are very interesting as they nicely show that the grain size has a large influence on the phase transition. Nevertheless, I am missing the transition temperature for the small grain sizes as a comparison. But as a literature value is given I do not insist on determining it for these studies.

Comment for further studies:

It would be interesting to produce something between 32 and 214 nm to investigate if the transition temperature continuously decreases with decreasing grain size or if there might be a threshold value at which the transition temperature “jumps” to significantly higher values.

Author Response

Dear Reviewer #1

We submit a revised manuscript of our paper titled “Grain size and phase transformation behavior of TiNi shape-memory-alloy thin film under different deposition conditions” (Manuscript ID: materials-856490). We have modified the manuscript accordingly, and the detailed answers are listed below point by point.

Reviewer 2 Report

The manuscript deals with the study of TiNi shape memory alloy thin films. The authors present how the deposition conditions are effect on the grain size of the films.

Although the dependency of grain size on deposition conditions has been well studied, I think this is a valuable work, could be in interest of the readers. The manuscript is good edited and contains a number of valuable experimental results, but it must be improved in some point. I recommend extending the introduction and discussion part. Add more information about the significance of shape memory alloys, what are the main parameters of these materials, etc. For example, the authors show the values of Ms, Rs, and As as transition temperatures at the end of the conclusion. What are these temperatures, why it is important to know them?

It is also recommended to check the English language style.

Author Response

Dear Reviewer #2

We submit a revised manuscript of our paper titled “Grain size and phase transformation behavior of TiNi shape-memory-alloy thin film under different deposition conditions” (Manuscipt ID: materials-856490). We have modified the manuscript accordingly, and the detailed answers are listed below point by point.

Reviewer 3 Report

I did not find any remarks concerning the content of the article. Mainly, on this article my remarks have a formal character.

  1. The text of the article contains some spelling errors. For example, "crystallizaing" (Page 1, line 37), Page 4, line 113, and so on.
  2. Most likely, owing to an error of the type fonts the Greek letter "mu" used in dimention of film thickness is replaced with the Latin letter "u". In order to avoid the similar technical failures it should be better to use the term "micron".

Author Response

Dear Reviewer #3

We submit a revised manuscript of our paper titled “Grain size and phase transformation behavior of TiNi shape-memory-alloy thin film under different deposition conditions” (Manuscript ID: materials-856490). We have modified the manuscript accordingly, and the detailed answers are listed below point by point.

Reviewer 4 Report

This paper describes grain size and phase transformation behavior of TiNi shape-memory-alloy thin film under different deposition conditions. The authors report that the thin film having grain size less than 35 nm, grown at 723 K, exhibited B2 structure through XRD measurements at all measured temperatures. These results indicated that a martensite phase transformation does not occur when the crystallized thin film is cooled to 183 K, which is consistent with the DSC results. The authors also found that the thin film having grain size greater than 200 nm, grown at 773 K, showed the two-step phase transformation process of B2 to R and subsequently to B19'. These experimental results are interesting and valuable.

  1. In line 37: crystallizaing => crystallizing
  2. All “um” should be written as “μm”.
  3. In line 86: The authors state that “The grain size increased as growth rate increased up to 38.00 nm/min, beyond which no increase in grain size was observed (Fig. 2(f)).”. Why did the grain size not show the increase beyond the growth rate of 38.00 nm/min?
  4. In line 104: minimizse => minimize
  5. In Fig. 3 (e): μm => μm
  6. In line 146: cnosistent => consistent
  7. In line 156: undego => undergo
  8. Line 166: The authors claims that “However, the phase transition temperature dropped below 123 K due to the small grain sizes.”. However, there is no evidence of phase transformation in Fig. 1 (c), Fig. 5 (a) and Fig. 6 (a) in the thin films grown at low substrate temperatures since the authors performed the DSC measurements within the range from 423 K to 123 K. Why did the authors not performed the DSC and XRD measurements at lower temperatures? The temperature of 123 K is the lowest value available in the equipment? The authors should use appropriate descriptions.

Author Response

Dear Reviewer #4

We submit a revised manuscript of our paper titled “Grain size and phase transformation behavior of TiNi shape-memory-alloy thin film under different deposition conditions” (Manuscript ID: materials-856490). We have modified the manuscript accordingly, and the detailed answers are listed below point by point.

Round 2

Reviewer 1 Report

Thanks to the Authors for the changes. I think the manuscript has significantly improved. I have just one minor comment:

The explanation about how the grain size is determined should also be added to the "Characterization of Thin Films" Section. The authors explained it to me in the response letter but it should also be explained to the readers of the paper.

If this minor change is made i am happy to accept the manuscript for publication.

Author Response

Thanks for your kindly review. We have modified the manuscript as your comment.
